# Microstructure and Properties of Aluminum–Graphene–SiC Matrix Composites after Friction Stir Processing

**DOI:** 10.3390/ma17050979

**Published:** 2024-02-20

**Authors:** Chen Wang, Xianyong Zhu, Yuexiang Fan, Jiaan Liu, Liangwen Xie, Cheng Jiang, Xiong Xiao, Peng Wu, Xiangmi You

**Affiliations:** 1School of Mechanical and Aerospace Engineering, Jilin University, Changchun 130022, China; chenwangowen@163.com (C.W.); fyx0524@163.com (Y.F.); xlw1126@163.com (L.X.); xiaoxiong20@mails.jlu.edu.cn (X.X.); 2Chongqing Research Institute, Jilin University, Chongqing 401123, China; 3College of Materials Science and Engineering, Jilin University, Changchun 130022, China; liuja@jlu.edu.cn; 4Changchun Baoze Technology Co., Ltd., Changchun 130051, China; bzwupeng@163.com; 5CISDI Chongqing Iron & Steelmaking Plant Integration Co., Ltd., No.11 Huijin Road North New Zone, Chongqing 401122, China; xiangmi.you@cisdi.com.cn

**Keywords:** friction stir processing, aluminum matrix composites, silicon carbide, graphene nanosheets, microstructure

## Abstract

Enhancing the mechanical properties of conventional ceramic particles-reinforced aluminum (Al 1060) metal matrix composites (AMCs) with lower detrimental phases is difficult. In this research work, AMCs are reinforced with graphene nanosheet (GNS) and hybrid reinforcement (GNS combined with 20% SiC, synthesized by shift-speed ball milling (SSBM), and further fabricated by two-pass friction stir processing (FSP). The effect of GNS content and the addition of SiC on the microstructure and mechanical properties of AMCs are studied. The microstructure, elemental, and phase composition of the developed composite are examined using SEM, EDS, and XRD techniques, respectively. Mechanical properties such as hardness, wear, and tensile strength are analyzed. The experimental results show that the GNS and the SiC are fairly distributed in the Al matrix via SSBM, which is beneficial for the mechanical properties of the composites. The maximum tensile strength of the composites is approximately 171.3 MPa in AMCs reinforced by hybrid reinforcements. The tensile strength of the GNS/Al composites increases when the GNS content increases from 0 to 1%, but then reduces with the further increase in GNS content. The hardness increases by 2.3%, 24.9%, 28.9%, and 41.8% when the Al 1060 is reinforced with 0.5, 1, 2% GNS, and a hybrid of SiC and GNS, respectively. The SiC provides further enhancement of the hardness of AMCs reinforced by GNS. The coefficient of friction decreases by about 7%, 13%, and 17% with the reinforcement of 0.5, 1, and 2% GNS, respectively. Hybrid reinforcement has the lowest friction coefficient (0.41). The decreasing friction coefficient contributes to the self-lubrication of GNSs, the reduction in the contact area with the substrate, and the load-bearing ability of ceramic particles. According to this study, the strengthening mechanisms of the composites may be due to thermal mismatch, grain refinement, and Orowan looping. In summary, such hybrid reinforcements effectively improve the mechanical and tribological properties of the composites.

## 1. Introduction

AMCs are a class of materials that include aluminum as the matrix and reinforcement materials embedded within it. These composites show better mechanical and other properties provided by the reinforcements than traditional alloys. Al 1060 has good processability to manufacture lightweight components, but low strength. It is practical to make it into AMCs to improve their mechanical properties. On the one hand, the uniform dispersion of reinforcement and reduced content of deleterious phase are challenging when fabricating AMC by Al 1060. On the other hand, high content and structural integrity of reinforcement are novel strategies that can further enhance the properties of AMCs. FSP is a relatively new surface modification technique derived from friction stir welding. Besides modifying the surface, the rotating tool causes severe plastic deformation (SPD) during FSP, contributing to homogenization and grain refinement. Moreover, SPD also provides additional strain hardening in the AMCs. The processing temperature of FSP is always below the aluminum alloy’s melting point, which can reduce the generation of Al oxides or carbides [1,2]. Therefore, FSP has the potential to fabricate metal matrix composites, especially in aluminum. However, the fabrication performance largely depends on the reinforcement characteristics and volume [3,4].

In the previous study, FSPed AMCs enhanced by ceramic particles such as SiC particles improved by 50% compared to the base metal in hardness and were 2.5 times higher in yield strength [2]. This improvement may be attributed to the grain refinement and uniform distribution of fragmented SiC particles caused by FSPed. 

However, the composites may also exhibit a decrement in ductility and toughness attributed to the deformable characteristics of ceramic reinforcements. Graphene (Gr is one of the thinnest and hardest known materials in the carbon group, which has extremely high mechanical properties (Young’s modulus: 1100 MPa; breaking strength: 130 GPa) high electrical conductivity, and high thermal conductivity. Therefore, Gr will be an ideal two-dimensional enhancement phase for fabricating AMCs. As for carbon nanotubes (CNTs), they agglomerate more easily than Gr because of their high aspect ratio and proneness to curling during the processing. Furthermore, Gr can also achieve superior tribological properties with a very small amount of content, which provides a lubricating effect and then reduces the friction and wear of the surface. GNS is more suited to initial experimental exploration than single-layer Gr and CNT because it easier to obtain and lower cost.

Due to Gr’s intrinsic van Der Waals force, it is easy to aggregate in the metal matrix, which may reduce the strength efficiency of Gr in AMCs [5]. Uniform dispersion of the reinforcement phase in the matrix is crucial to achieving high-property AMC fabrication. 

Thus, the challenges of using graphene as reinforcement may come down to poor dispersion, low interfacial bonding, carbide formation, and low structural integrity. In recent years, many investigations have focused on the pretreatment and fabrication methods of GNS-reinforced metal composites. For example, Kambiz et al. found that high-energy ball milling embedded GNS within the aluminum matrix to achieve more dispersal than low-energy mixing and ultrasonication [6]. Patil et al. found that metal matrix composites that utilized dry milling exhibited more uniform GNS dispersion and relatively higher yield strength than those that utilized solution ball milling. Their results also show that coating and bonding between the Gr and Al are better above 200 rpm [7]. Zhong et al. reported improvement of composites pretreated by SSBM, which showed improvements in tensile strength and elongation without Al_3_C_4_ because of the good structure and homogeneous dispersion of GNS, especially for high-content GNS [8]. Even though high-energy ball milling results in severe plastic deformation, it may also involve thermal effects and crystal defects. However, low-speed ball milling may cause insufficient refinement and uneven distribution. Compared to the first two methods, shift-speed ball milling can achieve uniform dispersion, fine grain, and structural integrity. 

Most of the previous research has paid more attention to temperature above melting point to develop structures with good mechanical properties. Diptikanta et al. and P. C. Mishra fabricated 20 wt.% SiC-reinforced Al 7075 using a stir casting process. The research indicated a significant enhancement in hardness [9,10]. Swain et al. dispersed nanoparticle SiC in molten Al to fabricate Al-SiCp nanocomposite material using a high-frequency mechanical vibrator, which might reduce the clustering of nanoparticles [11]. Shalini et al. attempted to utilize 12% nano SiC to reinforce pure Al by a high-energy ball milling process combined with electrical discharge machining. The use of nanoparticles significantly improved the mechanical properties [12]. Ravinat et al. utilized stir casting combined with heat treatment to fabricate Al-Si-Mg alloy reinforced with 10 wt% of Al_2_O_3_ particles. The decrease in wear rate may have been related to the addition of ceramic particles and oxide layers formed by heat treatment [13]. Those studies focused more on using powder fabrication methods to realize a finer grain and uniform mixing. The molding process also dispersed more homogeneously and minimized the detrimental phase by a sample procedure below the melt temperature. 

However, few investigations have studied the effect of size and content of hybrid reinforcement processed by shift-speed ball milling combined with FSP. This work focuses on the balance of strength and ductility of Al 1060 matrix composites reinforced with GNS and SiC. We take advantage of the plastic deformation caused by FSP, uniform dispersion realized by ball milling, and strengthening mechanism provided by reinforcements such as GNS and SiC. Finally, in our work, the minimization of the content of the detrimental phase, grain refinement and uniform dispersion of reinforcement is achieved. The optimal content of GNS in the hybrid reinforcement is obtained from this work. Combined with recent research on AMCs reinforced by different sizes of SiC, the hybrid reinforcement fills in a groove whose width is larger than the pin diameter of FSP. In addition, the influence of FSPed composites reinforced by different content of particles on microstructure, strength, hardness, and wear performance is investigated and discussed. The test results show that SiC and GNS distribute uniformly in combination with aluminum powder in the mixed powder without the generation of oxide or brittle hard phase Al_3_C_4_. In a series of experiments, an optimum scale and particle addition of the composites with high mechanical properties are explored.

## 2. Materials and Methods

In this study, Al-1060 plates of 200 mm in length, 75 mm in width, and 8 mm thickness with the necessary chemical composition and mechanical properties are shown in Table 1 and Table 2, respectively. Grooves 6.5 mm deep and 8 mm wide were machined along the center line of the plates and filled with mixed powder, which included Al powder (99.85% purity, 3 μm in diameter, Gaokexc, Beijing, China) mixed with GNS(99.5% purity purity,  ~ 5 μm in diameter and ~10 nm in thickness, XFNANO, Nanjing, China)and SiC (99.9% purity, Gaokexc, Beijing, China) combined with GNS, respectively. The GNS-reinforced composites were reinforced with varying concentrations (0.5, 1, 2 wt%). Meanwhile, SiC with average particle sizes of 8 μm was used to reinforce AMCs and the content was chosen as 20%, according to our preliminary exploration. The powders were mixed to prepare mixed powders with different contents of GNS and GNS combined with SiC by using a shift-speed planetary ball mill (Nanjing Boyuntong Instrument QM-3SP4, Nanjing, China).

All the powders before the ball milling were analyzed by scanning electron microscopy (SEM, JSM-7900F, Tokyo, Japan) to study their morphology. Figure 1a,b show the polarizing microscope-based metallographic structure and average grain size distribution counted by Nano-Measure 1.2 of Al 1060-H16. Figure 2a,b show the SEM-based micrographs and average grain size distribution of Al powder, whereas Figure 3a,b show the XRD and average spherical particle size distribution of Al powder. The micromorphology, EDS, and XRD are displayed in Figure 4 and Figure 5, respectively.

To improve the dispersion content of SiC and GNS in the matrix, the study used zirconia balls with diameters of 3 mm and 6 mm at a ratio of 1:3. The mixed powder with a ball–powder ratio of 8:1 was sealed in an agate jar filled with Ar. The shift-speed ball milling process was implemented in two steps: Firstly, it was ball milled at 120 rpm for 6 h. Secondly, it was placed in a high-speed mill at a rotation speed of 250 rpm for 2 h after 12 h static cooling. The micromorphology of the mixed powder and its element distribution were evaluated by SEM with EDS. XRD analysis was applied to scan the phases of the powder before and after mixing, determine the phases in the powder, and confirm no oxide or brittle hard phase Al_4_C_3_ were generated after ball milling. The schematic diagram of the shift-speed ball milling process is shown in Figure 6.

In the AMC preparation stage, we added the obtained mixed powder to the aluminum plate groove mold and compressed it. Then, we placed an aluminum plate with a thickness of 2 mm on it to prevent powder from flying during processing. A pinless tool was initially employed to compress the plate cover after it was filled with mixed powder in the groove to prevent the particles from scattering during FSP. The process parameters employed included tool rotational speeds of 2300 rpm in the first two passes and 1600 rpm in the last passes, a travel speed of 40 mm/min, a press amount of −0.4 mm, and an angle of 2.5°. The process parameters were selected based on trial experiments and previous research conducted by the author. The processing tool was made of H13 Tool Steel, which has a right-hand thread, and the profile was the cylinder. Other dimensional features such as length, pin diameter, and shoulder diameter were 5.8 mm, 7 mm, and 17 mm, respectively. The diameter of the pinless tool was 16 mm. The schematic representations of the friction stir welding process and overview of the pinless tool and pin are shown in Figure 7.

The microstructural characterization samples were prepared using grinding paper from 400 to 3000 grit and metallographically polished with 1 μm SiO_2_, then subsequently ultrasonically cleaned and etched using Keller reagent. The tensile strength was tested via the transverse uniaxial tensile test by an electronic universal testing machine (INSTRON INSTRON-5869, Norwood, MA, USA) at the rate of 1 × 10^−3^ s^−1^ three times. Moreover, the Vickers hardness was measured by a microhardness tester (HUAYIN HVS-1000A, Laizhou, China) at 100 g load applied for 10 s. The tribological properties of the processed surface were assessed by the dry sliding wear test conducted through a ball on a high-speed reciprocating friction testing machine (Zhongkekaihua, HSR-2M, Lanzhou, China) in the air at room temperature. The measured surfaces of the samples were initially smoothened by manual paper polishing to remove the surface asperities created during processing. The applied normal load was 10 N at a speed of 50 mm/min and was applied for 30 min to create a linear sliding distance equal to 15 m. The wear test specimens were 10 mm in length, 5 mm in width, and 6 mm in height. Tests were carried out with varying loads at 10 N and in straight lines back and forth at 50 mm/min for 30 min.

The wear rate (W, mm^3^/(Nm)) of specimens may be defined by considering the applied load (P, N) and sliding distance (L, m) as follows [14]:W = ΔV/(L × P)(1)

The worn surface’s volume (ΔV) may be calculated by the ratio of mass loss (Δm) to the apparent density (ρ) of the specimen, as follows:ΔV = Δm/ρ(2)

The devices of ball milling, FSP and the mentioned test are shown in Figure 8, and the above-mentioned samples are shown in Figure 9.

## 3. Results

### 3.1. Microscopic Morphology of Mixed Powders of Different Types and Ratios

From Figure 10, it can be seen that there was no significant change in the phase composition of the powder before and after ball milling, which preliminarily proves the rationality of the shift-speed ball milling process parameters. Meanwhile, the crystal phase of SiC in hybrid reinforcement is further confirmed by XRD. Figure 11a clearly shows that GNS is surrounded by Al powder, which further demonstrates that the low-speed ball milling process parameters used in this study are suitable for uniform mixing between particles and two-dimensional reinforcements with aluminum powder. Figure 11b shows GNS embedded in Al powder. The “particle clusters” tightly bonded with GNS and Al powder can be seen in Figure 11a. This illustrates the effectiveness of a close combination of GNS and aluminum powder by high-speed ball milling.

Figure 12 and Figure 13 show the SEM image and EDS mapping analysis of the powder after the aluminum powder is mixed with different contents of GNS and 20% SiC combined with 1% GNS through a shift-speed ball milling process. The captions only provide the reinforcement content; the remaining content is Al powder.

Through comparison, it can be found that the distribution of mixed powder containing 20% SiC + 1% GNS reinforcement is more uniform, and the Al powder is tightly wrapped. When the GNS content increases to 2%, agglomeration can be seen in the mixed powder (white arrow). The clustering of GNSs may be due to van der Waals force between layers, which can negatively affect the mechanical properties of AMCs.

### 3.2. Microstructure

As usual, the friction stir processing zone is divided into a nugget zone, a thermo-mechanically affected zone, a heat-affected zone, and base material. For convenience, the sample is divided into Up Zone (UZ), Research Zone (RZ), Down Zone (DZ), Base Material (BM), Advancing Side (AS) and Retreating Side (RS) in the study. These zones are shown in Figure 14. Figure 15a–c show the metallographic structure in AS, RZ, and RS reinforced by 100% Al powder, respectively. All of them show a smaller size than the base metal. According to the comparison, the grains in RZ experienced more refinement than the other two aspects and formed fine equiaxed grains. Because of the heat generated by two sources (one being the friction between the tool and the powder, with the other being the plastic flow of materials) dynamic recrystallization happens in the center of RZ. Meanwhile, the grains are broken by tool rotation. The other AMC samples were therefore only prepared using RZ. The performance and microstructural characterization measured in this article only apply to the red box (RZ). The mean grain size of AMCs is shown in Table 3. As stated in Table 3, a slight decrease in the grain size was observed when GNS were introduced. A more pronounced decrease in the grain size (4.9 μm) was observed when GNS combined with a high content of SiC were used. The change in the grain size was less significant when the GNS was present. This may be related to the GNS having broken down and provided nucleation sites during the intense deformation that occurred during ball milling and FSP. According to the previous study, particle stimulation nucleation takes place during FSP [2].

Figure 16 clearly shows the polarized metallographic structure of RZ in an AMC reinforced by different GNS content. Most of the grains shown here are relatively coarse, with some relatively fine grains, which have rolling characteristics, distributed between them.

Figure 17 displays the metallographic structure of RZ in AMC reinforced by the combination of 20% SiC with 1% GNS. As can be seen in the Figures, the microstructure grain is fine and the SiC particle size is refined. But a small amount of SiC agglomerates in the boundary.

Figure 18 and Figure 19 show the SEM image and EDS mapping analysis of the fabricated surface composite reinforced by hybrid reinforcement, Al powder and different GNS content. It can be found that both GNS and SiC are uniformly dispersed in the base metal. The composite zone is mainly composed of aluminum, followed by oxygen. The oxygen may be incorporated into FSP. The oxidation of the aluminum matrix occurs at high temperatures. As can be seen in Figure 19a,b, the carbon elements are more evenly distributed in the composite zone with the increase in GNS content, indicating that GNSs are more evenly dispersed in the composite zone by FSP. Meanwhile, the accumulation of GNS can be observed in Figure 19c. GNSs tend to form clusters in the substrate when the content continues to increase. According to the EDS in Figure 18a, some parts of SiC and Al turn to silicon dioxide and alumina oxide during this process, which may be attributed to the enhancement of the mechanical strength of the AMCs.

The EDS maps also show the presence of reinforced particles in the SZ, which also confirms the nearly uniform dispersal of reinforcement in all AMCs. Homogeneous dispersion is mainly attributed to the effect of ball milling and FSP.

AMC is fabricated by FSP. The processing parameters, reinforced particle type, size, and content all affect the metallographic morphology, the size of the processed composite, the uniformity, and the particle size of the reinforcement distribution. Changes in grain size and morphology at the center of RZ after processing may be attributed to the following reasons: Firstly, the frictional heat and severe plastic deformation generated during the FSP process cause sufficient dynamic recrystallization. Secondly, SiC and two-dimensional GNSs have the effect of pinning grain boundaries to prevent grain growth, and their high thermal conductivities promote rapid temperature reduction in the processing area. Moreover, the tool breaks the coarse grains during the rotation. 

The differences in grain size and morphology at the center of RZ are mainly caused by the difference in the thermal conductivity of the composite material after processing. Thermal conductivity is mainly affected by the type and content of reinforcements [15,16].

The uniform distribution of reinforcement in aluminum matrix composites is mainly determined by the stirring effect of the tool. On the one hand, when the material is in a plastic state, the high-speed rotating stirring tool drives the reinforcement to flow and breaks the reinforcement grains. The reinforcement is uniformly distributed in the “onion ring” shape at the center of the RZ [7,17]. On the other hand, collisions also occur between reinforcements and aluminum powder during the tool rotation. Therefore, the reinforcement is tightly bonded to the aluminum powder and provides the power to reduce the size of the reinforcement [18].

### 3.3. Tensile Strength

To obtain the optimal proportion of reinforcement under the optimal mechanical properties, different GNS contents in reinforcing particles and hybrid reinforcement are investigated. The specific tensile characteristic values are shown in Table 4. The AMCs reinforced by 0.5, 1, 2%GNP, and hybrid reinforcement show an increase in ultimate tensile stress of~34.4%, ~34.6%, ~30.3%, and ~90.3%, respectively, as compared to as-received Al 1060 alloy. And the elongation increases by ~16.2%, ~32.4%, ~−2.1%, and ~30.0%, respectively. The increase in Al-GNS is attributed to the combined effect of grain refinement during FSP. The tensile properties first increase and then decrease with the increase in GNS content, as excessive content can lead to reinforcement agglomeration. Moreover, the bonding force between layers is lower in GNS. If the tensile direction goes against the GNS, the premature initiation of fracture cracks may be born at the interlayer of GNS. Therefore, too much GNS can induce more cracks and reduce tensile properties. 

Figure 20 shows tensile curves of Al 1060 and AMC reinforced by Al, SiC, and GNS, respectively. Their fracture morphology is shown in Figure 21. According to the figure, it can be found that regardless of the content, the UST value of the AMC reinforced by GNS is greater than that of the substrate and 100% aluminum powder aluminum matrix composite material because FSP can crush particles [19,20].

When the GNS content is 1%, the tensile mechanical properties of aluminum matrix composites are the best. While considering the UST value, the EI value is also taken into consideration. We found that 1%GNS obtains the highest value for both the EI values and UTS among the GNS/Al composites. There are numerous deep and large dimples in the fracture surface of 0.5% and 1% GNS/Al composites, which indicates plastic fracture. The dimples become more uniform and finer along with the increase in GNSs. However, when the GNS content exceeds 2%, it is difficult to disperse homogeneously. The cluster of GNS only appears in 2% of GNS/Al composite (white arrow in Figure 21), which may relate to the reduction in tensile strength.

The following experimental results can be used to fabricate a hybrid aluminum matrix composite material of GNS and SiC. The AMC is prepared by combining 20% SiC with the maximum UTS and 1% GNS with the maximum EI. The aim is to obtain an aluminum matrix composite material with the best UST and EI, which overcomes the shortcomings of reinforced particles.

In recent studies, four major strengthening mechanisms affected the tensile properties of aluminum matrix composites: fine-grained strengthening, dispersion strengthening, thermal mismatch stress strengthening, and load transfer strengthening [16,21,22].

The grain size of the material is refined after processing, resulting in an improvement in the mechanical properties, especially with regard to tensile stress. The refinement of grains and increased grain boundaries cause stronger resistance to dislocation motion. The Hall–Petch equation describes the strengthening mechanism of the grain boundary. The reinforcement in the matrix combined with the homogeneous distribution significantly enhanced the mechanical properties of the AMCs. In general, the coefficients of thermal expansion of aluminum, GNS, and SiC particles are 21.4 × 10^−6^/°C, −6 × 10^−6^/°C and 4 × 10^−6^/°C, respectively. It can be seen that the difference in the thermal expansion coefficient between GNS toward aluminum and SiC toward aluminum is small, so thermal mismatch stress strengthening is not the main influencing mechanism of tensile properties. GNS are two-dimensional reinforcements with various anisotropies. The large specific surface area of GNS led to good interfacial adhesion between them and the matrix, which improved the mechanical properties of AMCs reinforced by GNS and hybrid reinforcements. However, when the content of GNS was too high, a large-scale aggregation structure could be formed in the metal. High concentrations of graphene may also introduce defects and cracks, further weakening the tensile properties of the material. On the one hand, load transfer strengthening may be attributed to the existence of the micron-sized particles. The load transfers from the soft Al 1060 base metal, across the interface between the matrix and reinforcement, to the harder micro-reinforcement particles. Meanwhile, the harder micro-reinforcements can withstand most of the external stresses to strengthen the base metal. On the other hand, the addition of nano-reinforcement can hinder the passing of dislocations and thus promote Orowan loops around the particles. When the tensile direction is in line with the optimal performance direction, the particles can bear a large load, resulting in a large EI of GNS in aluminum matrix composites. This is consistent with previous research results [23]. As for hybrid reinforcement, the SiC can lead to inhomogeneous local deformation and occupy the preferential grain boundaries, which can promote further refinement of the grain. In summary, this article will combine SiC with GNS to utilize both load transfer strengthening and dispersion strengthening. The noteworthy feature is the increase in UTS of AMCs reinforced by hybrid reinforcement by 31.8% in this study as compared to the UTS obtained by Khodabakhshi et al. [24], who used the same process without the SiC content and mixed method. Ultimately, we optimize both the UST and the EI of SiC/GNS aluminum matrix composites.

### 3.4. Hardness

In this section, the hardness of each sample is investigated. The results are shown in Figure 22 and Table 5. As compared to composite reinforced by 100% Al, AMCs reinforced by 0.5, 1 and 2% GNS and hybrid particles show 2.3%, 24.9%, 28.9%, and 41.8% increases in hardness. This result indicates that the addition of GNS has positively influenced the composites’ hardness, as has the base matrix. Due to its high hardness, GNS has encountered indenters and restricted the indentation, resulting in improved hardness. According to the test results, 1. as the content of GNS increases, the microhardness increases. The improved hardness of Al-GNS composites is related to the wonderful mechanical properties of GNS, which provide high restraining force for deformation during indentations. This enhancement in hardness may also be attributed to the strengthening mechanism and refined microstructure, which can be proved by the microstructure mentioned above. 2. The effect of different GNS content on the microhardness value of AMC is not as large as that of hybrid reinforcement, due to the GNS content being very low. The variation amplitude of microhardness values at the center of RZ is smaller than that of AS and RS, and the variation amplitude on both sides is very small. The fluctuation range is determined by the uniformity of the reinforcement distribution. The increased trend of hardness reinforced by GNS is similar to the research work conducted by Manjunath et al. [25].

As for the AMCs reinforced by hybrid reinforcement, the hardness shows a 10.1% enhancement compared to the AMCs only reinforced by the same SiC content [26]. According to Hall–Petch theory, hardness is inversely proportional to grain size. GNS and fine SiC particles disperse in the processed zones, providing more nucleation sites for the re-precipitation of new grains and more precipitates in the aluminum matrix [22]. The composite reinforced by hybrid reinforcement shows the highest microhardness value (67.8 ± 1.8 HV). Both SiC particles and GNS can withstand the load, but SiC as a hard ceramic particle is more capable of withstanding large loads than GNS under the same vertical force, according to the previous study [23,27]. Also, SiC can occupy the preferential grain boundaries to exhibit geometrically necessary dislocation. The geometrically necessary dislocation can restrict dislocations in the composites to resist deformation [28,29]. This is also attributed to the grain refinement during FSP, the abrasive behavior of SiC, load-carrying characterization, and the restriction of dislocation motion by SiC in the matrix. Moreover, the mismatch in CTE among the reinforcement (GNS (~1.0 × 10^−6^/K) and SiC (~4.02 × 10^−6^/K)) and the matrix (Al (~23.6 × 10^−6^/K)) causes the matrix to form an excess of geometrically required dislocations adjacent to the reinforcement/matrix interface during composite production Therefore, it can be concluded that the exfoliation and uniform distribution of GNS, hardness of SiC particle, and CET mismatch between GNS, SiC, and Al improve the microhardness of SiC/GNS composites.

### 3.5. Friction and Wear Performance

Figure 23 illustrates the time-variant friction coefficient of Al 1060 and AMCs reinforced by Al powder and different GNS content, as well as the average friction coefficient (COFave). As shown in the picture, the average friction coefficient of GNS aluminum matrix composites varies with their content. COFave decreases with the increment of the GNS content. This may be attributed to the following factors: firstly, GNS has a two-dimensional area and its movable layered structure has lower adhesion to the surface [30]; secondly, its higher specific surface serves as a film to spread rather than gather in the pothole. Moreover, the wrinkled and folded morphology can dissipate the stress. 

The wear surface of the AMCs is depicted in Figure 23. As observed from the worn surface morphologies, there are parallel grooves on the worn surface along the sliding direction. The presence of grooves corresponds to the abrasion component of the wear mechanism. Therefore, the abrasive wear mechanism may be described as the main wear mechanism. 

The COFave variance of composites reinforced by 2% GNS is lower than other contents due to the characteristic of GNS. The wear test results also proved that GNS was a good solid lubricator due to the graphene’s lubricating property. As for AMCs reinforced by hybrid reinforcement, the addition of SiC resulted in a decrease in the wear rate due to the hard ceramic structure and uniform distribution of high SiC content. The high SiC content serves as a load-bearing element to restrict the subsurface damage prior to sliding and forms a protective layer of lubricating film to prevent direct contact between the base metal and steel counterface. Moreover, Archard theory indicates the opposite relationship between the wear rate and the hardness of the material [14]. According to the hardness and wear test results, higher hardness induced enhancement of the wear resistance of AMC reinforced by hybrid reinforcement. 

In summary, the average friction coefficient value of the prepared aluminum matrix composite material is related to its microhardness, and the fluctuation range of the friction coefficient over time is related to the uniformity of reinforcement and tissue distribution. As SiC is a hard particle, its average friction coefficient is much higher than that of graphene, but SiC has excellent wear resistance. Therefore, adding GNS to decrease the high average friction coefficient of SiC is an ideal strategy to enhance the wear resistance and decrease the COFave of composites.

In Figure 23, the COFave of AMC reinforced by 100% aluminum powder is relatively large, while the COFave of aluminum composites containing GNS is small, even lower than the substrate. Moreover, adding GNS to SiC particles can reduce COFave due to the solid lubrication effect of GNS. The observed patterns conform to the previous literature [31,32]. The shadow on the surface reveals the tribolayer by the shearing of GNS caused by plastic deformation. This tribolayer has previously been reported to impede direct contact between the friction pair, which can effectively decrease COF and the wear rate [33]. 

Figure 24 shows worn surface micrographs of AMC. Regardless of the content of GNS, AMCs show low wear volume without pits. Therefore, the dominant wear mechanism of AMC reinforced by GNS is abrasive wear accompanied by oxidation wear. This indicates that GNS plays a key role in solid lubrication and forms an excellent interface with Al to improve the load-bearing capacity of AMCs [34]. Figure 24e displays typical characteristics of abrasive wear on AMC reinforced by hybrid reinforcement in the form of numerous furrows and scratches of varying depths. Therefore, the core wear mechanism is abrasive wear. SiC improves the wear behavior of AMCs due to its self-lubricating mechanism. In addition to the load carrying of SiC, the protrusion of SiC also decreases the real contact area between the rubbing surfaces. This phenomenon favors the formation of a stable and protective surface layer on the wear surface and reduces the wear rate. Due to the following factors, the friction coefficient of hybrid composites is much lower than that of AMCs reinforced by GNS.

## 4. Conclusions

In this study, the Al 1060 composites reinforced by various contents of GNS and hybrid reinforcements were fabricated through shift-speed ball milling, followed by FSP technology. The conclusions are listed as follows: Shift-speed ball technology contributed to grain refinement, homogeneous dispersion of reinforcement, and exfoliation of GNS. As the GNS content increased from 0 to 2%, the tensile strength of the composites first increased and then decreased due to the agglomeration of GNS. The composites containing 1% of GNSs had the highest tensile strength. The continuous increment of microhardness was achieved when the GNS increased. Hybrid reinforcements combining SiC particles and GNS enhanced the mechanical properties of the composites. The addition of hybrid reinforcements with 20% SiC and 1% GNS resulted in a significant improvement in tensile strength. The improved effect of hybrid reinforcements was larger than single GNS reinforcement. As for microhardness, reinforcing the surface by 0.5, 1, 2% GNS and a hybrid of reinforcement increased the hardness by 2.3%, 24.9%, 28.9%, and 41.8%, respectively. The increment was related to the dispersion strengthening of the SiC and GNS, grain refinement, and thermal mismatches between Al and SiC, GNS. Consistent with the hardness improvements, the friction coefficient decreased with the increase in the GNS content. The coefficient of friction decreased by approximately 22% and 43% with the reinforcement of 0.5, 1, 2% GNS, respectively. Additionally, 2% GNSs had the lowest friction coefficient (0.409). SiC is highly wear-resistant as a hard particle, while GNS is not only wear-resistant but also lubricated. Compared to 1% GNS, the combination of 20% SiC with 1% GNS showed greater improvement than single reinforcement in terms of wear resistance and lubrication. The improved wear resistance of AMC reinforced by hybrid reinforcement is attributed to the self-lubrication of GNS, reduction in the contact area with substrate, and the greater load bearing of ceramic particles. The abrasion is identified as the main wear mechanism in the AMCs.

Overall, the research on aluminum–graphene–SiC matrix composites offers promising improvements for other characteristics (conductivity and corrosion resistance). Further research is needed to optimize the properties of these materials and develop cost-effective manufacturing processes.

## Figures and Tables

**Figure 1 materials-17-00979-f001:**
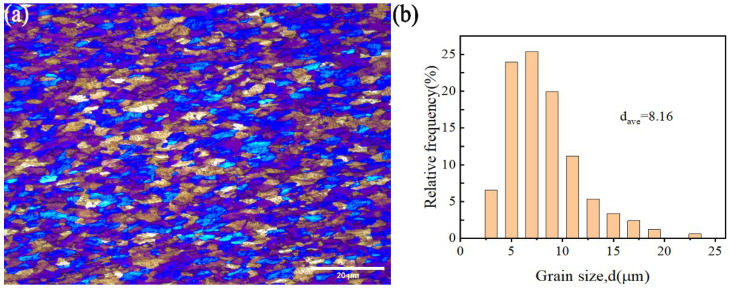
(**a**) Metallographic structure of Al 1060-H16; (**b**) average particle size distribution.

**Figure 2 materials-17-00979-f002:**
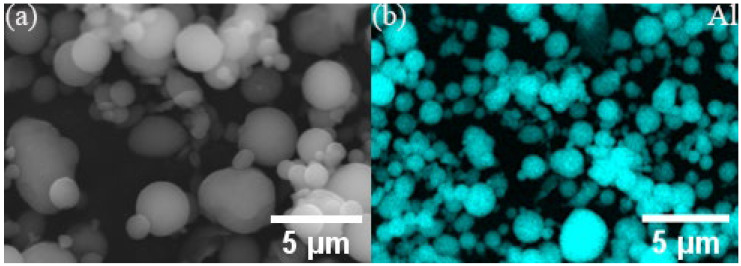
(**a**) Micromorphology and (**b**) EDS of Al powder.

**Figure 3 materials-17-00979-f003:**
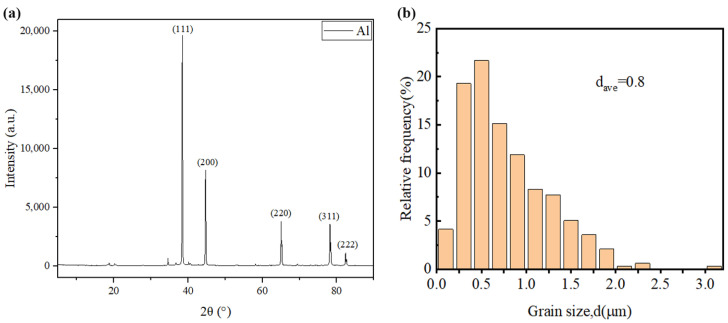
(**a**) XRD of Al powder and (**b**) average spherical particle size distribution.

**Figure 4 materials-17-00979-f004:**
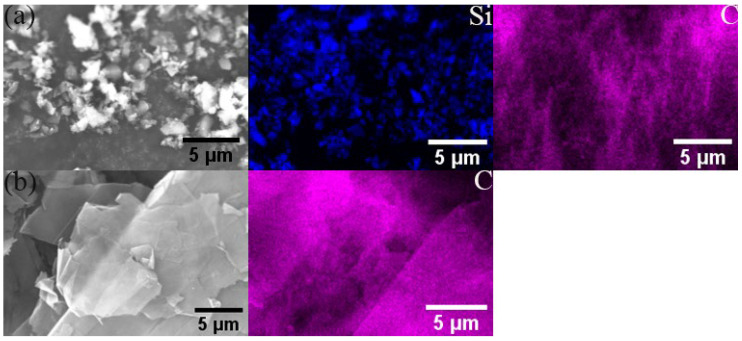
Micromorphology and EDS of (**a**) SiC, (**b**) GNS.

**Figure 5 materials-17-00979-f005:**
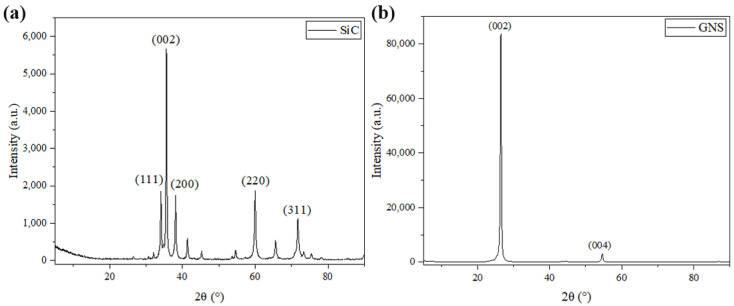
XRD of (**a**) SiC, (**b**) GNS.

**Figure 6 materials-17-00979-f006:**
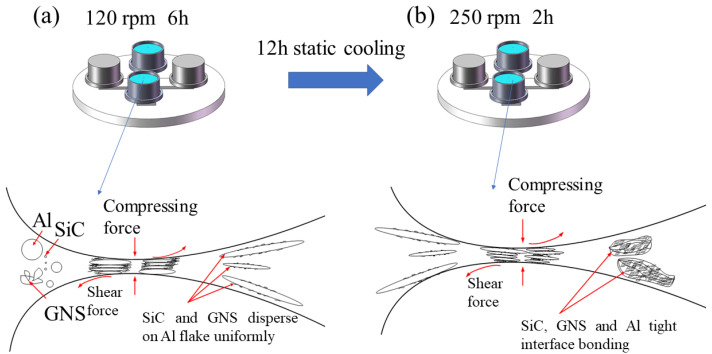
The schematic diagram of the shift-speed ball milling process (**a**) first step of ball milling (**b**) second step of ball milling after 12 h static cooling.

**Figure 7 materials-17-00979-f007:**
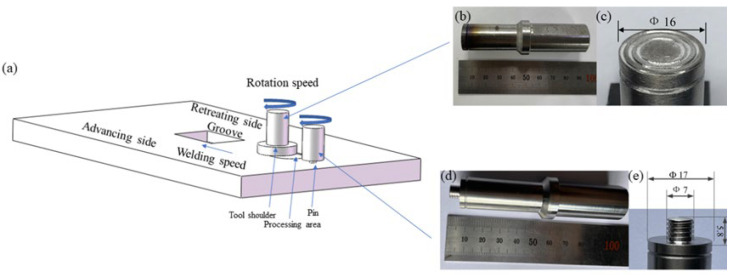
(**a**) Schematic representations of friction stir welding process and overview of (**b**) pinless tool and its geometric dimensions (**c**), (**d**) pin tool and (**e**) its geometric dimensions.

**Figure 8 materials-17-00979-f008:**
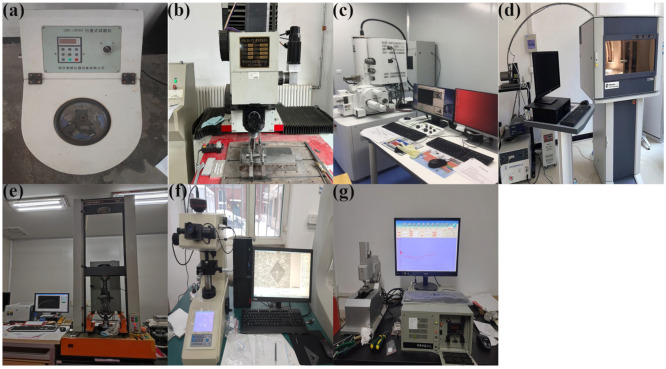
Experimental setup of (**a**) ball milling (**b**) friction stir processing and the devices used to test (**c**) SEM, (**d**) XRD, (**e**) tensile, (**f**) friction and (**g**) microhardness, respectively.

**Figure 9 materials-17-00979-f009:**
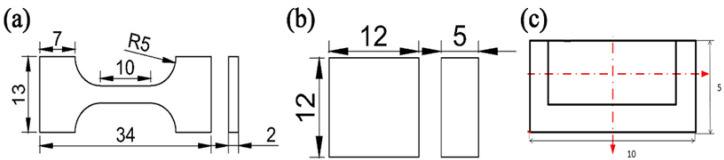
Schematic illustration of the measurement samples for fabricated composites (**a**) tensile specimen size, (**b**) friction and wear specimen size, and (**c**) microhardness sample.

**Figure 10 materials-17-00979-f010:**
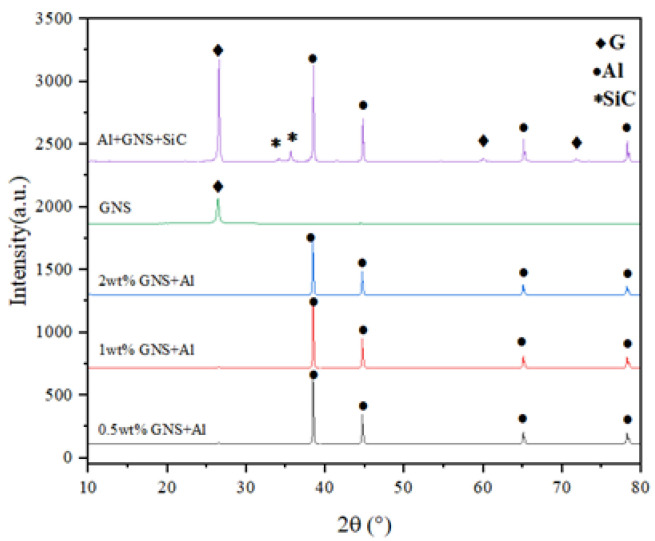
XRD phase composition before and after shift-speed ball milling.

**Figure 11 materials-17-00979-f011:**
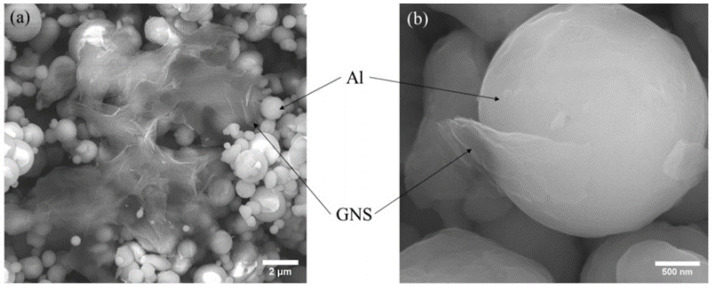
Micromorphology of GNS/Al powder (**a**) during low-speed ball milling, (**b**) during high-speed ball milling.

**Figure 12 materials-17-00979-f012:**
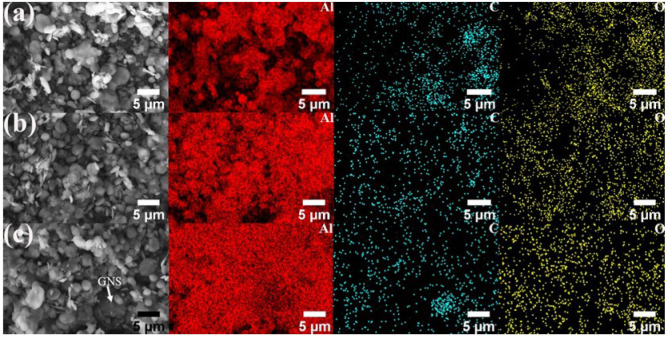
SEM image and EDS mapping analysis of mixed powder: (**a**) 0.5%, (**b**) 1%, (**c**) 2%GNS.

**Figure 13 materials-17-00979-f013:**
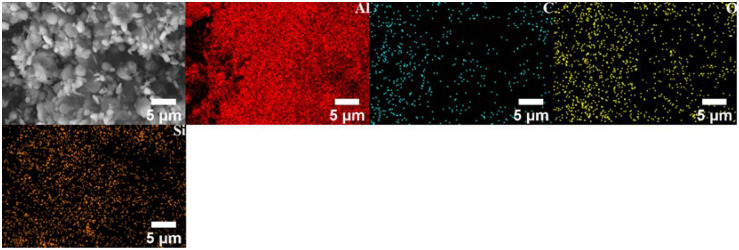
SEM image and EDS mapping analysis of mixed powder: 20% SiC + 1% GNS.

**Figure 14 materials-17-00979-f014:**
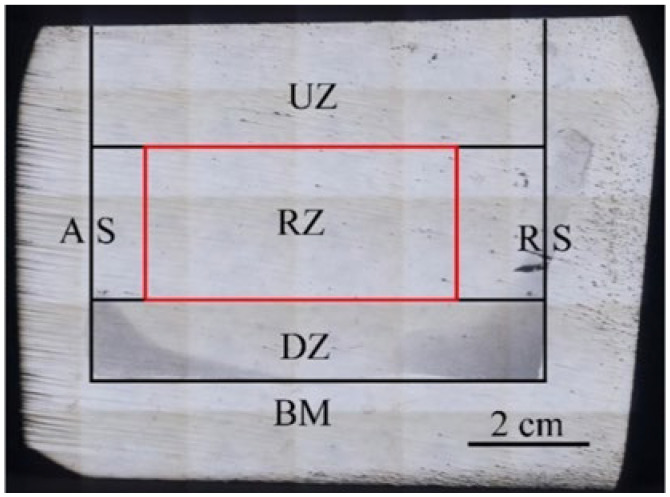
Micromorphology zoning for machining cross-section.

**Figure 15 materials-17-00979-f015:**
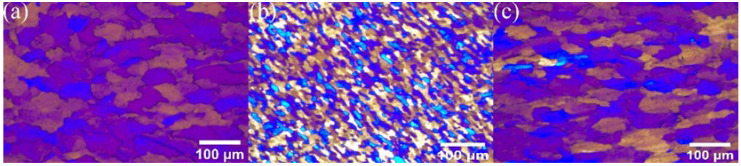
Metallographic structure of (**a**) AS, (**b**) RZ, (**c**) RS of 100% aluminum powder-reinforced AMC.

**Figure 16 materials-17-00979-f016:**
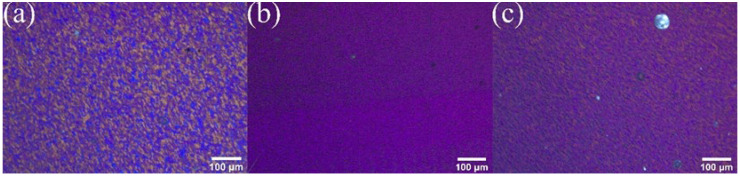
Metallographic structure of RZ in (**a**) 0.5% (**b**) 1% (**c**) 2% GNS-reinforced AMC.

**Figure 17 materials-17-00979-f017:**
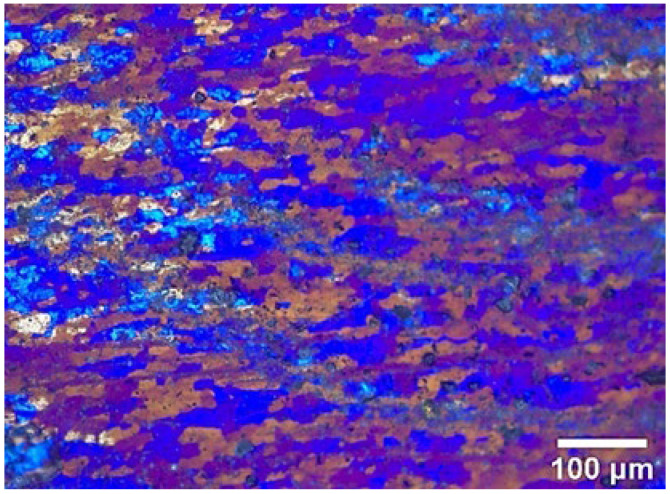
The metallographic structure of RZ in AMC reinforced by the combination of 20% SiC with 1% GNS.

**Figure 18 materials-17-00979-f018:**
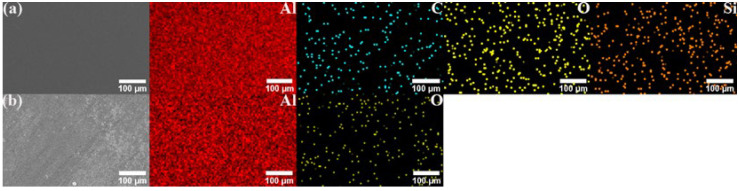
SEM image and EDS mapping analysis of the SZ of the fabricated surface composite reinforced by (**a**) 20% SiC + 1% GNS and (**b**) 100% Al powder.

**Figure 19 materials-17-00979-f019:**
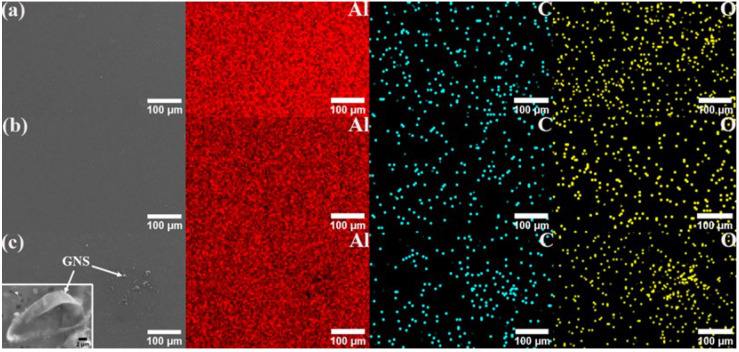
EDS mapping analysis of the SZ of the fabricated surface composite reinforced by (**a**) 0.5%, (**b**) 1%, and (**c**) 2% GNS, respectively.

**Figure 20 materials-17-00979-f020:**
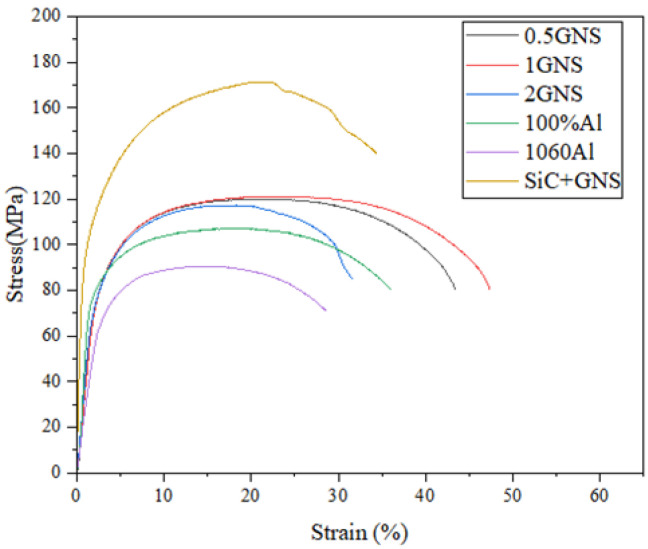
Tensile curves of Al 1060 and AMC.

**Figure 21 materials-17-00979-f021:**
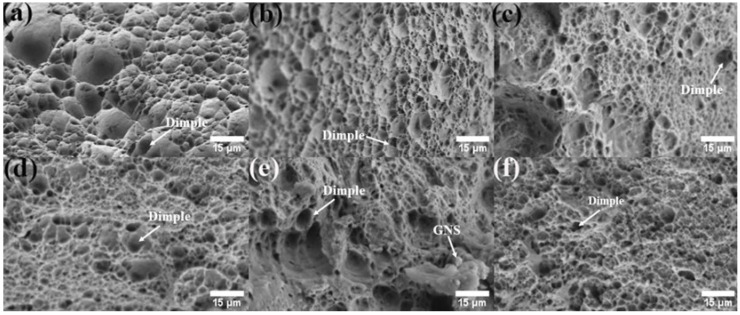
The fracture morphology of (**a**) Al 1060, AMC reinforced by (**b**) Al powder, (**c**–**e**) by 0.5, 1 and 2% GNS, (**f**) 20% SiC + 1% GNS.

**Figure 22 materials-17-00979-f022:**
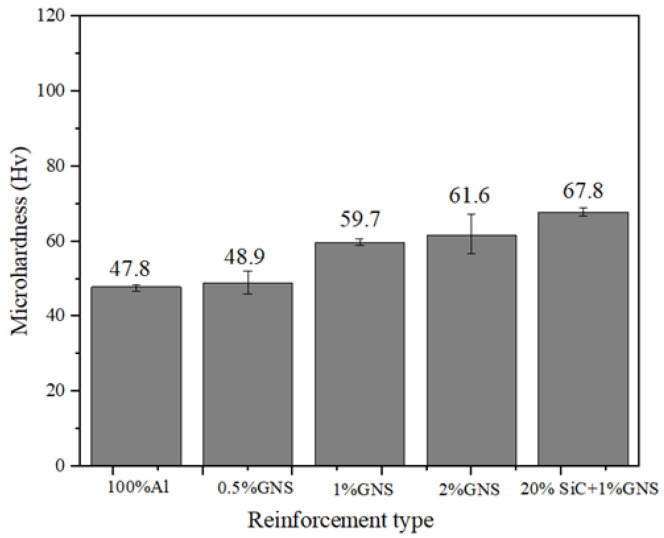
Hardness curves of AMC reinforced by Al powder, 0.5, 1 and 2% GNS, and 20% SiC + 1% GNS.

**Figure 23 materials-17-00979-f023:**
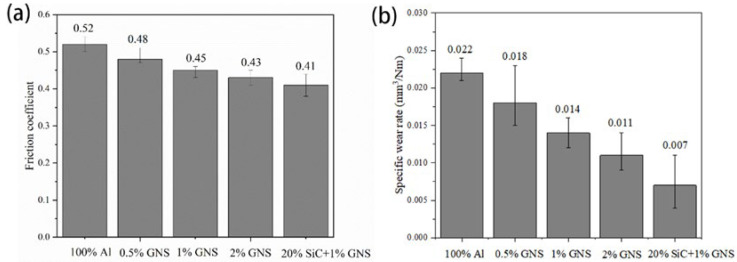
(**a**) Friction coefficient and (**b**) specific wear rate of AMCs reinforced by Al powder, 0.5, 1 and 2% GNS and 20% SiC + 1% GNS.

**Figure 24 materials-17-00979-f024:**
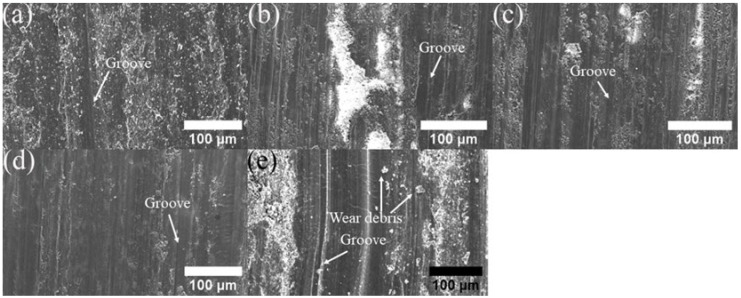
SEM worn surface micrographs of AMC reinforced by (**a**) Al powder, (**b**–**d**) by 0.5, 1 and 2% GNS, (**e**) 20% SiC + 1% GNS.

**Table 1 materials-17-00979-t001:** Chemical composition of Al 1060-H16 (wt. %).

Materials	Mg	Cu	V	Zn	Mn	Si	Fe	Ti	Al
Al 1060	0.03	0.05	0.05	0.05	0.03	0.25	0.35	0.03	Bal.

**Table 2 materials-17-00979-t002:** Mechanical properties of 1060-H16 Al.

Material	YSσ_0.2_/MPa	UTS/MPa	El/%	Hardness/HV
Al 1060	60	90	42	30

**Table 3 materials-17-00979-t003:** The mean grain size of AMCs.

Material	100% Al	0.5% GNS	1% GNS	2% GNS	1% GNS + 20% SiC
Average grain size (μm)	20.2 ± 3.3	9.3 ± 1.5	6.6 ± 0.5	11.4 ± 0.8	4.9 ± 1

**Table 4 materials-17-00979-t004:** Tensile mechanical properties of base metal and AMCs.

Material	YSσ_0.2_/MPa	UTS/MPa	El/%
Al 1060	60	90	42
100% Al powder	70.7	107	47
0.5% GNS	71.1	121	48.8
1% GNS	71.2	121.1	55.6
2% GNS	67.4	117.3	41.1
20% SiC + 1% GNS	42.4	171.3	50.8

**Table 5 materials-17-00979-t005:** The hardness value of the different areas in the base metal and AMCs.

Material	RZ of AS/HV ^a^	RZ/HV ^b^	RZ of RS/HV ^c^
Al 1060	30 ± 0.2	30 ± 0.2	30 ± 0.2
100%Al powder	47.8 ± 0.2	47.8 ± 0.2	47.8 ± 0.2
0.5% GNS	48.9 ± 3.9	48.9 ± 3.1	48.9 ± 5.1
1% GNS	59.7 ± 1.3	59.7 ± 0.9	59.7 ± 2
2% GNS	61.6 ± 7	61.6 ± 5.6	61.6 ± 9.9
20% SiC + 1% GNS	67.8 ± 1.8	67.8 ± 1.1	67.8 ± 2.3

^a^ RZ of AS: researcher zone of advancing side, ^b^ RZ: researcher zone, ^c^ RZ of RS: researcher zone of retreating side.

## Data Availability

Data are contained within the article.

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
