# Peer review of "Microstructure and Properties of Aluminum–Graphene–SiC Matrix Composites after Friction Stir Processing"

_materials, 2024, doi:10.3390/ma17050979_

Round 1
Reviewer 1 Report (Previous Reviewer 4)
Comments and Suggestions for Authors
Dear Authors
I have read the revised article submitted to materials journals. I appreciate that you have considered my comments and improved the paper. But still, I need some clarification regarding the manuscript. My comments are as follows
The introduction is too lengthy. Could you make it short? The abstract can be of 250 - 300 words.
The authors are requested to read this article and cite them appropriately https://doi.org/10.1016/j.ijlmm.2023.01.002
Why did the authors add 20wt% of SiC, which is usually not recommendable due to the forming of agglomeration? How was the reinforcement wt % fixed for the test
Fig. 5 and Fig.3 peak needs to be indexed.
Meanwhile, SiC reinforced composites 142 with average particle sizes of 8μm and the content was chosen as 20% according to our 143 former studies. - Kindly cite these sentences
For microstructure, what etchant did the author use?
What is the grain size of the developed sample
What was the load used for hardness? Please also mention the parameters used for wear testing. How did u calculate the wear rate? what was the repeatability of the test? I would like to know the repeat of experiments for tensile and hardness.
The wear mechanism should be discussed in the manuscript. Kindly do that
Why did the YS reduce, but UTS and elongation increase in Table 3 after 2wt% GNS? In fact, how could you produce tensile samples from ball mill products? could you please provide the image of the tensile sample used?
Conclusion I would suggest the authors make into points which will be helpful
Author Response
Thank you for your encouraging remarks and valuable comments. We corrected our text according to your comments in the attached Word file. The writing style especially the discussion has been modified. Our responses to your comments are as follows:
Comment 1: The introduction is too lengthy. Could you make it short? The abstract can be of 250 - 300 words.
Response to C1: We thank the reviewer for pointing this out! The content in the abstract and introduction has been modified and deleted(L13-20,42-44,53-56,68-70,83-101).
Comment 2: The authors are requested to read this article and cite them appropriately https://doi.org/10.1016/j.ijlmm.2023.01.002
Response to C2: Thanks for your useful suggestions! We have carefully read the literature you recommended. We find it very valuable for our discussion and investigation in the article. Therefore, the literature was cited and the article was improved. If you have more literature recommendations, please feel free to add them. We included reference in (13) Ravinath, H.; Ahammed I, I.; P, H.; Devan S, A.; Senan V R, A.; Shankar, K. V.; S, N. Impact of Aging Temperature on the Metallurgical and Dry Sliding Wear Behaviour of LM25 / Al2O3 Metal Matrix Composite for Potential Automotive Application. International Journal of Lightweight Materials and Manufacture 2023, 6 (3), 416–433. https://doi.org/10.1016/j.ijlmm.2023.01.002.
Comment 3: Why did the authors add 20wt% of SiC, which is usually not recommendable due to the forming of agglomeration? How was the reinforcement wt % fixed for the test
Response to C3: According to our previous attempt, when the content of SiC reaches 20 weight percent, the mechanical properties, especially tensile strength have improved largely. Therefore, our research utilized 20% 8μm SiC to enhance further AMCs reinforced by GNS. The EDS and SEM of mixed powder show the homogeneous distribution of reinforcement.
Comment 4: Fig. 5 and Fig.3 peak needs to be indexed.
Response to C4: We thank the reviewer for pointing this out! The related peaks have been marked
Comment 5: Meanwhile, SiC reinforced composites 142 with average particle sizes of 8μm and the content was chosen as 20% according to our 143 former studies. - Kindly cite these sentences
Response to C5: We thank the reviewer for pointing this out! The 20% SiC is attributed to our preliminary exploration in the early stage.
Comment 6: What is the grain size of the developed sample
Response to C6: Thanks for your suggestion. The grain sizes of all samples have been added to the article(Table 3). The discussion of grain size was also added. (L250-256)
Comment 7: What was the load used for hardness? Please also mention the parameters used for wear testing.
Response to C7: Vickers hardness was measured by a microhardness tester at 100 g load applied for 10 s (L180-182).
Comment 8: For microstructure, what etchant did the author use?
Response to C8: The microstructural characterization samples were etched using Keller reagent.(L176-178)
Comment 9: How did u calculate the wear rate? what was the repeatability of the test? I would like to know the repeat of experiments for tensile and hardness.
Response to C9: the wear rate was calculated by the equation mentioned in reference 14 (L191-196). All the tests were conducted three times and the data was taken the average.
Comment 10: The wear mechanism should be discussed in the manuscript. Kindly do that
Response to C10: The wear rate and coefficient of friction were measured by a high-speed reciprocating friction testing machine. The applied normal load is 10 N at a speed of 50mm/min and conducted for 30 min making a linear sliding distance equal to 15 m. (L184-198)
Comment 11: Why did the YS reduce, but UTS and elongation increase in Table 3 after 2wt% GNS? In fact, how could you produce tensile samples from ball mill products? could you please provide the image of the tensile sample used?
Response to C11: The YS reduced, but UTS and elongation increased may be attributed to the small part cluster of GNS. The tensile samples were fabricated by FSP and cut by wire cutting. The SEM of tensile fracture is shown in fig 21.
If any other areas need improvement, please feel free to contact us at any time. Wishing you a happy Chinese New Year.

Reviewer 2 Report (Previous Reviewer 2)
Comments and Suggestions for Authors
This paper deals with ¨ Microstructure and Properties of Aluminum-Graphene-SiC 2 Matrix Composites by Friction Stir Processing¨. The manuscript topic is interesting, and from this reviewer's point of view, this article is not acceptable for publication in its present form.
1- The abstract is confusing. It is hard to follow the author's approach in their research. Please add the exact methodology and more quantities of results.
2- Many typos and grammatical errors can be detected in the manuscript.
3- The aim and scope of this paper are not described. The author should state what kind of problem they want to solve in this research and what the novelty of this paper is.
4- Please remove a bunch of citations.
5- The quality of images is low.
6- The introduction should be improved. The first part of the introduction should be removed.
7- Scientific discussions on the results are low.
Mentioned Above.
Author Response
Thank you for the helpful comments, which allowed us to improve our manuscript. We made the revisions accordingly. The main corrections and the responses to your comments follow:
Comment 1: The abstract is confusing. It is hard to follow the author's approach in their research. Please add the exact methodology and more quantities of results.
Response to C1: We thank the reviewer for pointing this out! The content in the abstract has been modified, especially for the approach in the study and quantities of results (L13-20,28-31)
Comment 2: Many typos and grammatical errors can be detected in the manuscript.
Response to C2: We feel very sorry for the flaws and imprecise data. Thanks for pointing it out. According to your comments, we have modified the content and increased the experiments.
Comment 3: The aim and scope of this paper are not described. The author should state what kind of problem they want to solve in this research and what the novelty of this paper is.
Response to C3: We feel great thanks for your professional review work on our article. The previous investigations focused less on AMC fabricated by Al 1060 towards FSP. In this article, we utilized GNS and the high content of SiC to enhance the properties. The uniform dispersion of reinforcement and less content of deleterious phase are the main problems to be solved. Therefore, shift speed ball milling combined with FSP is the novelty in our study.(L42-45)
Comment 4: Please remove a bunch of citations.
Response to C4: According to your suggestion, we have reconsidered and removed some references.
Comment 5: The quality of images is low.
Response to C5: Thank you for your helpful comments. Your suggestion really means a lot to us! We have checked the images in the whole article and adjusted Fig4,11,12 and 14 appropriately. If any figure leaves you unsatisfied, please point out and contact us.
Comment 6: The introduction should be improved. The first part of the introduction should be removed.
Response to C6: We thank the reviewer for pointing this out! Taking into account the opinions of various experts, the first part of the introduction has been modified and deleted some content. (L42-44)
Comment 7: Scientific discussions on the results are low.
Response to C7: Your suggestion really means a lot to us! We fully agree with the expansion of the discussion on the tensile, hardness and wear(L367-369,384-391,411-418,427-435). The conclusion also made corresponding modifications.
If any other areas need improvement, please feel free to contact us at any time. Wishing you a happy Chinese New Year.

Reviewer 3 Report (Previous Reviewer 3)
Comments and Suggestions for Authors
The paper has been revised and may be accepted.
Author Response
Thank you for your careful review. If you have any further suggestions, please feel free to contact us at any time. Wishing you a happy Chinese New Year.

Reviewer 4 Report (Previous Reviewer 1)
Comments and Suggestions for Authors
I have thoroughly reviewed the manuscript, along with the revisions and the authors' responses to my previous comments. I appreciate the efforts made by the authors to address the concerns raised during the initial review. After careful consideration, I am pleased to confirm that the revisions have significantly improved the manuscript. In my opinion, the work is now suitable for publication in Materials.
Author Response
Thank you for your careful review. If you have any further suggestions, please feel free to contact us at any time. Wishing you a happy Chinese New Year.

Reviewer 5 Report (New Reviewer)
Comments and Suggestions for Authors
The presented study entitled "Microstructure and Properties of Aluminum-Graphene-SiC Matrix Composites by Friction Stir Processing" aims to create a graphene nanosheet (GNS) and aluminum reinforced with silicon carbide (SiC). The study analyzed the influence of GNS content on the microstructure and mechanical properties of GNS/Al. Furthermore, within the study, the authors analyzed the effect of adding SiC on the mechanical properties of GNS and SiC/Al composites. The submitted manuscript represents an interesting study, logically organized. In the following text, I will comment on individual parts of the manuscript.
Abstract: the abstract contains all the essential information that represents the study as a whole. From the definition of the basic problem, through the proposal of its solution, the solution procedure and the basic results. It is quite long, which is not the standard scope of an abstract, but I have no comments on its content. Due to the large number of abbreviations and symbols used, I would consider creating a list of them before the actual text of the manuscript. It is problematic and complicated to search for abbreviations and their meaning in the text. Creating a list of abbreviations and symbols would make reading the text easier.
1. Introduction: this part presents a literary overview of the problem. I agree with the used literary sources and the method of their processing. I have no comments on this part.
2. Materials and Methods: in this chapter, the authors analyze in detail the material used, the way the experiment was carried out, as well as the measurement methodology. The form of treatment of this chapter meets the condition of possible reproducibility of the experiment. However, I have a comment about Fig. 1b and 3a.
- Since the average is strongly dependent on the type of data distribution and the presence of outliers, and at the same time the images shown show significant skewness (skew coefficient > 0), it is necessary to verify the data in the graph about the average value of the grain diameter.
3. Results: this chapter is divided into separate parts according to the analyzed characteristics (Microscopic morphology of mixed powders of different types and ratios; Microstructure; Tensile Strength; Hardness). This division makes the text easier to follow. Individual parts are described in an interesting and concise way. I have no comment on that. But the relevance of the described results is rather questionable. Each measurement must be understood as a random variable. Every measurement is burdened, in the sense of measurement theory, with gross, random and systematic error. For that:
- Above all, Fig. 22 and Fig.23 is defined and described in the first plan. Based on the above, the differences in values (Hardness; Friction coefficient) must be described by relevant statistical tests of the difference in mean values. Although at first glance the value of 0.52 may appear to be higher than the value of 0.48 (Figure 23a) considering the standard deviation, the values may be the same in a statistical sense, and the actual value of the difference may just be a coincidence.
- Therefore, I recommend subjecting the measurement results of the mentioned parameters to statistical methods to clearly confirm that the differences depending on the "type of tested material" are caused by the actual material used. The results will thus be given a relevant weight and their evaluation will not be so dependent on subjective evaluation.
4. Conclusion – it is a summary of the entire study and I have no fundamental comments about this chapter
But what I miss is a critical discussion of the obtained results. I expect a critical comparison of the results obtained by the authors with other relevant studies. Fill in please.
Overall, I rate the study as very interesting. My aim is to point out certain shortcomings so that the authors can improve their study. I do not want the authors to perceive my comments as criticism of their work. After incorporating my comments, I recommend publishing the study in the journal Materials.
Author Response
We appreciate your detailed analysis and specific suggestions. Based on your opinion, we have referred to previous studies and modified the article.
Comment 1: Creating a list of abbreviations and symbols would make reading the text easier.
Response to C1: Your suggestion really means a lot to us! Abbreviations appearing only once have been deleted. The abbreviations also concentrate on the abstract and the beginning of the introduction.
Comment 2: It is necessary to verify the data in the graph about the average value of the grain diameter.
Response to C2: The average value of the grain diameter has been checked by imagej and the average grain size of AMCs has also been shown in table3.
Comment 3: Above all, Fig. 22 and Fig.23 is defined and described in the first plan. The values may be the same in a statistical sense, and the actual value of the difference may just be a coincidence. The results will thus be given a relevant weight and their evaluation will not be so dependent on subjective evaluation.
Response to C3: The data shown in Fig22 and 23 are average value for three times of tests. Three same types of samples were measured to calculate the average value in each test. The error bar represents the difference in the average value of each test. A bar chart is more convincing than an error bar. Therefore, AMCS reinforced by hybrid reinforcements is higher than AMCs reinforced by 2%GNS. The result also conforms to the previous studies cited in the references. In general, hardness is directly proportional to wear resistance. Moreover, the wear rate can be more reflective of wear resistance according to the studies. Combined with the coefficient of friction, the wear resistance of AMCs can be evaluated comprehensively.
Comment 4: Conclusion – it is a summary of the entire study and I have no fundamental comments about this chapter
But what I miss is a critical discussion of the obtained results. I expect a critical comparison of the results obtained by the authors with other relevant studies. Fill in please.
Response to C4: We feel great thanks for your professional review work on our article. The result of tensile strength and hardness in our study has been compared to relevant previous studies. (L386-388, 413-414)
If any other areas need improvement, please feel free to contact us at any time. Wishing you a happy Chinese New Year.

Round 2
Reviewer 1 Report (Previous Reviewer 4)
Comments and Suggestions for Authors
Accept in current form
Comments on the Quality of English Languagefine
Reviewer 2 Report (Previous Reviewer 2)
Comments and Suggestions for Authors
The authors answered the comments properly.
This manuscript is a resubmission of an earlier submission. The following is a list of the peer review reports and author responses from that submission.
Round 1
Reviewer 1 Report
Comments and Suggestions for Authors
Title: Well-relate to the presented work.
This work presents a study on the fabrication of nanocarbon-reinforced aluminum matrix composites using a shift speed ball milling technology and friction stir process (FSP). The research investigates the distribution of graphene nanosheets (GNS) and silicon carbide (SiC) powder in the aluminum (Al) matrix, focusing on their impact on the mechanical properties of the resulting composites.
Introduction:
AMC abbreviation is not properly introduced.
GNS is not properly introduced in the main text.
Need to revise the literature as many of the references are being lump into, without a proper structure.
Materials:
What is 8μm SiC? Name of sample? Surely the size in not uniformly 8μm in all.
Figure 4: The text is too small, needs to be revised. Also, for some past in Figure 5.
Conclusion should be written in a paragraph.
Overall, other than the mentioned above, writing style also needs to be improved a lot. Before the manuscript can be considered.
Comments on the Quality of English LanguageAs mentioned in the above comments.
Reviewer 2 Report
Comments and Suggestions for Authors
This paper deals with the ¨ Microstructure and Properties of Aluminum-Graphene-SiC Matrix Composites by Friction Stir Processing¨. The manuscript topic is interesting, but many flaws can be detected in this article. From this reviewer's point of view, this form of article is not acceptable for publication, and it needs serious and major revisions:
1- The abstract is confusing. It is hard to follow the author's approach in their research. Please add the exact methodology and more quantities of results.
2- Many typos and grammatical errors can be detected in the manuscript.
3- The aim and scope of this paper are not described. The author should state what kind of problem they want to solve in this research and what is the novelty of this paper.
4- EDS or XRD analysis for used powder in Fig. 2 and Fig. 3 is necessary.
5- The quality of images, especially SEM, is low. Please add small spaces between SEM images to be better visualized. The scale bars are blurred (Fig. 17 and Fig. 20).
6- Discussion on the results is weak. Please improve it.
Reviewer 3 Report
Comments and Suggestions for Authors
The paper is good but needs revision as follows:
1. Indicate the research gap after the literatutre review. Some references are recommended pertaining to the research and may be included.
(a) Research progress on nano-metal matrix composite (NMMC) fabrication method: A comprehensive review
(b) Experimental investigation into characterization and machining of Al+ SiCp nano-composites using coated carbide tool
(c) Tool wear and cutting force investigations during turning 15 wt% SiCp-Al 7075 metal matrix composite
(d) Study of machining characteristics of Al-SiCp12% composite in nano powder mixed dielectric electrical discharge machining using RSM
(e) Optimization of machining parameters and development of surface roughness models during turning Al-based metal matrix composite
(f) Multi-response optimization of process parameters using Taguchi method and grey relational analysis during turning AA 7075/SiC composite in dry and spray cooling environments
2. How can you ensure the uniform distribution of 1060Al matrix composites reinforced with GNS and SiC.
3. Provide the experimental/fabrication setup and measurement set up photograph in the paper.
4. Show the experimental setup for friction stir set up.
5. How parameters for friction stir set up has been made?
6. When the content of GNS is 1%, the tensile mechanical properties of aluminum matrix composites are the best. Explain with reasons behind it.
7. Conclusions should be supported by the data and write the future scope.
Reviewer 4 Report
Comments and Suggestions for Authors
My comments are attached herewith

Minor corrections